materials science/environmental chemistry/
inorganic chemistry

biochar, phosphate removal and recovery,
metal oxides, maximum adsorption capacity,
adsorption rate

**Author for correspondence:**
Shuhang Wang
e-mail: wangsh@craes.org.cn

This article has been edited by the Royal Society of Chemistry, including the commissioning, peer review process and editorial aspects up to the point of acceptance.

# A comparative study on phosphate removal from water using *Phragmites australis* biochars loaded with different metal oxides

Pengfei Wang[1], Mengmeng Zhi[2], Guannan Cui[3],
Zhaosheng Chu[1] and Shuhang Wang[1]

[1]National Engineering Laboratory for Lake Pollution Control and Ecological Restoration, State Environment Protection Key Laboratory for Lake Pollution Control, Chinese Research Academy of Environmental Sciences, Beijing 100012, People's Republic of China
[2]School of Environmental and Municipal Engineering, Xi'an University of Architecture and Technology, Xi'an 710055, People's Republic of China
[3]School of Ecology and Environment, Beijing Technology and Business University, Beijing 100048, People's Republic of China

PW, 0000-0002-8810-0297

Metal oxide-loaded biochars are a promising material to remove phosphate from polluted water to ultra-low concentrations. To facilitate preparing the metal oxide-loaded biochar with the best phosphate adsorption performance, five biochars loaded with Al, Ca, Fe, La and Mg oxides, respectively (Al-BC, Ca-BC, Fe-BC, La-BC and Mg-BC) were produced using *Phragmites australis* pretreated with 0.1 mol $AlCl_3$, $CaCl_2$, $FeCl_3$, $LaCl_3$ and $MgCl_2$, respectively, characterized, and phosphate adsorption kinetics and isotherms of the biochars were determined. The maximum phosphate adsorption capacities ($Q_m$) of the biochars ranked as Al-BC ($219.87 \, mg \, g^{-1}$) > Mg-BC ($112.45 \, mg \, g^{-1}$) > Ca-BC ($81.46 \, mg \, g^{-1}$) > Fe-BC ($46.61 \, mg \, g^{-1}$) > La-BC ($38.93 \, mg \, g^{-1}$). The time to reach the adsorption equilibrium ranked as La-BC (1 h) < Ca-BC (12 h) < Mg-BC (24 h) = Fe-BC (24 h) < Al-BC (greater than 72 h). $Q_m$ of Ca-BC, Fe-BC, La-BC and Mg-BC depend on the molar content of metals in the biochars. The small phosphate adsorption rate of Al-BC is due to the slow intra-particle diffusion of phosphate attributed to the undeveloped porosity and dispersed distribution of AlOOH crystals on the Al-BC surface. Mg-BC is suggested for phosphate removal from water considering adsorption rate and capacity. Al-BC is applicable when a long contact time is allowed, e.g. as a capping material to immobilize phosphate in lake sediments.

# 1. Introduction

In order to control the eutrophication of aquatic ecosystems, tremendous efforts have been made in wastewater treatment plants (WWTPs) to remove phosphorus (P). For example, the total P concentration in the effluent of municipal WWTPs should not exceed the strictest limit of $0.5 \, \text{mg l}^{-1}$ in China (GB18918-2002) and $1.0 \, \text{mg l}^{-1}$ in Europe (Council Directive 91/271/EEC). However, the eutrophication of the receiving water bodies may be aggravated even if the discharge limit is met, especially when the dilution effect of the receiving water body is low [1]. More strict discharge limits $(0.01–0.1 \, \text{mg l}^{-1})$ are or will be applied [2]. Moreover, purification of polluted water from non-point sources such as agricultural runoff and stormwater overflows has drawn increasing attention since P input from these non-point sources becomes important after the strict discharge limits have been met in WWTPs [3]. It is thus necessary to develop efficient and cost-effective P removal technologies to treat such polluted water with a low P concentration. Adsorption is a suitable technology for such purposes and has the advantages of simple design, easy operation and cost-effectiveness [4].

Biochar produced from waste biomass is an attractive adsorbent due to the large surface area, stability and presence of various surface functional groups [5,6]. Moreover, the P-adsorbed biochar can be reused as soil amendment and slow-release fertilizer to improve soil quality and productivity, enabling simultaneous P recovery and waste biomass reuse [7–9]. Nevertheless, the pristine biochar usually has a small or negative adsorption capacity for phosphate [10]. Impregnation of the biomass with metal salts before pyrolysis, among which Al [11–13], Ca [14,15], Fe [16,17], La [18,19] and Mg [20–22] salts are more commonly used, to produce modified biochars loaded with metal oxides, could increase the phosphate adsorption capacity of the biochar due to the good phosphate affinity of the metal oxides [23].

However, the biochar loaded with which metal oxide has the highest phosphate adsorption capacity and simultaneously a fast phosphate adsorption rate, and what are the factors determining the phosphate adsorption capacities and rates of the biochars loaded with different metal oxides, are important questions to be answered before using the modified biochars in practice. Existing studies generally use one or a mixture of two or three metal salts as the modification agent and a direct comparison of the results from existing studies is made impossible by the broad variation of biomass types, biochar preparation conditions (pyrolysis temperature, addition amount of metals, etc.) and phosphate adsorption conditions (pH, initial phosphate concentration, etc.) which greatly affect the phosphate adsorption characteristics of the modified biochars [11,22,24]. In a few studies [25,26], biochars loaded with different types of metal oxides were prepared using different metal salts as the biomass modification agents and the phosphate removal efficiencies of the biochars from aqueous solution or wastewater were compared without considering the phosphate adsorption kinetics and isotherms of the different biochars.

The purposes of this study are (i) to investigate and compare the characteristics and phosphate adsorption kinetics and capacities of the biochars loaded with different metal oxides, and (ii) to identify possible factors determining the phosphate adsorption capacities and rates of the biochars loaded with different metal oxides.

# 2. Material and methods

## 2.1. Biochar preparation

Straws of *Phragmites australis*, common reed, which is the most frequently used plant in constructed wetlands worldwide and needs to be regularly harvested and disposed of [27], was selected as the biomass feedstock for producing the biochars. *Phragmites australis*, harvested from a constructed wetland on the northern shore of Lake Erhai, Dali, China, was dried under the sun for more than one week. The straws were collected and cut into 5 cm pieces. After being washed with the tab water and rinsed with the deionized water, the straw pieces were dried at 105°C for 24 h. They were then crushed with a stainless crusher (CS-700, Cosuai, China) and passed through a 2 mm sieve. Twenty grams of reed straw powder and 0.1 mol analytical grade $AlCl_3$, $CaCl_2 \cdot 2H_2O$, $FeCl_3 \cdot 6H_2O$, $LaCl_3 \cdot 6H_2O$ and $MgCl_2 \cdot 6H_2O$, respectively, were added in 250 ml Erlenmeyer flasks with 200 ml deionized water. All chemicals used in this study were of analytical grade and purchased from Beijing Chemical Works, China. After being shaken in an oscillator (SHZ-28A, Haocheng, China) at 25°C and $150 \, \text{r min}^{-1}$ for 24 h, the mixture in each flask was transferred to a Petri dish, dried at 105°C to a

constant weight, and pyrolysed in a tube furnace (GSL 1200X-KQ-LT, Zhuochi, China) under $N_2$ atmosphere ($0.4\,l\,min^{-1}$). The temperature in the furnace was raised at $5°C\,min^{-1}$ from the room temperature to 600°C and maintained at 600°C for 2 h. The $N_2$ flow was started 30 min before the heating to exclude the air. After cooling to the room temperature under the $N_2$ flow, the obtained biochar was weighed and sieved with 0.25, 0.5 and 1.0 mm sieves. The fraction between 0.25 and 0.50 mm was used for the experiment. The biochars modified with the five metal chlorides were denoted Al-BC, Ca-BC, Fe-BC, La-BC and Mg-BC, respectively. Pristine biochar pyrolysed from the raw reed straw powder was also prepared under the identical conditions and denoted BC.

## 2.2. Phosphate adsorption experiments

Phosphate adsorption kinetic experiments with the six biochars were conducted with synthetic $KH_2PO_4$ solution with $20\,mg\,l^{-1}$ phosphate–phosphorus ($PO_4^{3-}-P$ ). One gram biochar and 500 ml $KH_2PO_4$ solution (biochar concentration $2\,g\,l^{-1}$), the pH of which was adjusted to 7.0 using $1.0\,mol\,l^{-1}$ HCl and NaOH solution, were added in 1 l Erlenmeyer flasks. After being sealed with Parafilms, the flasks were shaken in the oscillator at 25°C and $150\,r\,min^{-1}$ for 72 h. Thirty-five millilitre samples were taken from each flask at 0.5, 1, 1.5, 2, 3, 4, 6, 8, 12, 24, 48 and 72 h after the start of shaking, respectively, with syringes. The samples were settled for 5 min and the pH of the supernatant was measured. Afterwards, the supernatant was filtrated through 0.45 µm membrane filters and the filtrate was stored for analysis. Each kinetic experiment was conducted in triplicate, as was each isotherm experiment.

Phosphate adsorption isotherm experiments with the six biochars were conducted in 50 ml centrifuge tubes. 0.08 g biochar and 40 ml $KH_2PO_4$ solution (pH 7.0, biochar concentration $2\,g\,l^{-1}$) with 100–$1500\,mg\,l^{-1}$ $PO_4^{3-}-P$ were added in the tubes. After being shaken at 25°C and $150\,r\,min^{-1}$ for 24 h, the mixture in the tube was settled for 5 min. The pH of the supernatant was measured. Afterwards, the mixture was filtrated through 0.45 µm membrane filters and the filtrate was stored for analysis. In the experiments with the initial $PO_4^{3-}-P$ concentration of $1500\,mg\,l^{-1}$, the solids left on the filter were washed with 30 ml deionized water six times, dried at 105°C for 24 h and sealed in plastic bags for characterization. In the phosphate adsorption isotherm experiments with Al-BC, the shaking time was extended to 168 h because adsorption equilibrium was not reached within 72 h and the initial $PO_4^{3-}-P$ concentration was increased to $3000\,mg\,l^{-1}$ because the phosphate adsorption amount did not cease to increase in the initial $PO_4^{3-}-P$ concentration range of $100–1500\,mg\,l^{-1}$.

## 2.3. Biochar characterization and analysis methods

$PO_4^{3-}-P$ concentrations in the samples were analysed with a spectrophotometer (UV-1700, Shimadzu, Japan) using the ammonium molybdate spectrophotometric method (GB11893-89). The ash contents of the biochars were determined by heating the samples in a muffle furnace at 750°C for 6 h. The C, H and N contents of the biochars were determined with an elemental analyser (Vario MACRO cube, Elemental, Germany). The O content was approximated as the total mass of the biochar minus the ash content and the C, H and N contents [10]. The Al, Ca, Fe, La and Mg contents of the biochars were analysed with an inductively coupled plasma optical emission spectrometer (710-ES, Varian, USA) after microwave-assisted digestion. The points of zero charge ($pH_{pzc}$) of the biochars were determined with the same method as in Yin et al. [26]. The specific surface area and pore size distribution of the biochars were determined at 77 K through $N_2$ adsorption and desorption (3H-2000PS2, Beishide Instrument, China) using the Brunauer–Emmett–Teller (BET) method. The surface functional groups of the biochars were analysed with a Fourier transform infrared (FTIR) spectrometer (Nicolet is5, Thermo Scientific, USA). Surface morphologies of the biochars were observed with a scanning electron microscope (SU8010, Hitachi, Japan). Crystal compositions of the biochars were determined by an X-ray diffractometer (D8 Advance, Bruker, Germany) with Cu Kα radiation. Processing of the X-ray diffraction spectra was performed with the MDI Jade software (v. 6).

Fitting the kinetic and isotherm models to the experimental data was performed with Origin 2018 using the non-linear fitting method. Correlation between the phosphate adsorption capacity/rate and the physicochemical properties of the biochars was analysed with IBM SPSS Statistics 19 using the least squares method [26].

**Table 1.** Basic physicochemical properties of the biochars.

|  | Al-BC | Ca-BC | Fe-BC | La-BC | Mg-BC | BC |
|---|---|---|---|---|---|---|
| yield (%) | 40.38 | 55.82 | 55.91 | 58.23 | 37.27 | 29.98 |
| ash content (%) | 38.42 | 39.49 | 44.42 | 68.50 | 34.21 | 13.04 |
| C content (%) | 48.10 | 39.49 | 34.48 | 21.94 | 54.67 | 77.41 |
| H content (%) | 1.60 | 1.88 | 0.81 | 0.94 | 2.05 | 2.36 |
| N content (%) | 0.34 | 0.26 | 0.35 | 0.20 | 0.53 | 0.99 |
| O content (%) | 11.54 | 18.88 | 19.94 | 8.42 | 8.54 | 6.20 |
| Al content (%) | 15.52 | 0.001 | 0.01 | 0.001 | 0.04 | 0.01 |
| Ca content (%) | 0.03 | 17.99 | 0.07 | 0.01 | 0.03 | 0.09 |
| Fe content (%) | 0.04 | 0.02 | 22.34 | 0.01 | 0.02 | 0.34 |
| La content (%) | 0.0005 | 0.0003 | 0.01 | 38.58 | 0.0001 | 0.0003 |
| Mg content (%) | 0.05 | 0.08 | 0.03 | 0.03 | 16.18 | 0.14 |
| H/C molar ratio | 0.40 | 0.57 | 0.28 | 0.51 | 0.45 | 0.37 |
| O/C molar ratio | 0.18 | 0.36 | 0.43 | 0.29 | 0.12 | 0.06 |
| (O + N)/C molar ratio | 0.19 | 0.36 | 0.44 | 0.30 | 0.13 | 0.07 |
| $pH_{pzc}$ | 6.6 | 7.6 | 2.1 | 7.3 | 11.3 | 6.3 |

# 3. Results and discussion

## 3.1. Biochar properties

### 3.1.1. Yield, elemental composition and $pH_{pzc}$

Pretreatment of the biomass with the five metal chlorides increased the yield, ash content and Al, Ca, Fe, La and Mg metal content of the biochar from 29.98% to 37.27–58.23%, 13.04% to 34.21–68.50% and 0.57% to 15.64–38.63%, respectively, and decreased the C content of the biochar from 77.41% to 21.94–54.67% (table 1). The yield, ash content and metal content increase and C content decrease extents followed the same order as the molar mass of the metal elements in the added metal chlorides, i.e. La-BC > Fe-BC > Ca-BC > Al-BC > Mg-BC. Organic compounds of the biomass such as hemicellulose, cellulose and lignin were decomposed and the decomposition products were partly volatilized during the pyrolysis at 600°C [28]. The inclusion of metal chlorides in the biomass accelerated the biomass pyrolysis and reduced the volatilization rate [18], thereby increasing the biochar yield. Moreover, the metal chlorides decomposed into metal oxides which were impregnated in the biochar matrix (figure 2a) and hydrogen chloride which was volatilized [29]. The impregnation of metal oxides in the biochar resulted in increased metal content, yield and ash content of the biochar and a relative reduction of the carbon content [30]. The same molar amount of metal chlorides (0.1 mol) was added in the biomass for modification. Therefore, the metal contents, yields and ash contents of the modified biochars ranked in order of the molar mass of the metal elements and the C content ranked in the opposite order.

The modified biochars had much higher O/C and (O + N)/C molar ratios (0.12–0.43 and 0.13–0.44) than the pristine biochar (0.06 and 0.07), indicating a larger number of oxygen-containing functional groups and higher polarity and hydrophilicity [10], which facilitates the adsorption of polar contaminants such as phosphate. Except for Fe-BC, the modified biochars also had higher H/C molar ratios (0.40–0.57) than BC (0.37), suggesting a lower degree of carbonization and more plant organic residues in the modified biochars [31].

The $pH_{pzc}$ of the adsorbent is related to the electrostatic interactions between the adsorbent and adsorbate. When the solution pH is below the $pH_{pzc}$, the adsorbent possesses a net positive surface charge, enabling the electrostatic attraction of phosphate [32]. The $pH_{pzc}$ of the modified biochars except Fe-BC (6.6-11.3) were higher than that of BC (6.3), and the $pH_{pzc}$ of Mg-BC was the highest. This is due to the high $pH_{pzc}$ values of the metal oxides loaded on the biochars, especially MgO,

**Table 2.** Textural features of biochars.

| biochars | Al-BC | Ca-BC | Fe-BC | La-BC | Mg-BC | BC |
|---|---|---|---|---|---|---|
| $S_{BET}$ (m² g⁻¹) | 5.35 | 7.96 | 14.67 | 73.94 | 89.82 | 3.63 |
| $S_{micro}$ (m² g⁻¹) | 2.01 | 0 | 0 | 20.61 | 64.74 | 0 |
| $V_{pore}$ (×10⁻³ cm³ g⁻¹) | 35.60 | 34.10 | 58.20 | 94.20 | 98.30 | 27.70 |
| $V_{micro}$ (×10⁻³ cm³ g⁻¹) | 1.37 | 0 | 0 | 10.92 | 33.42 | 0 |
| $D_{pore}$ (nm) | 26.61 | 17.14 | 15.87 | 5.10 | 4.38 | 30.53 |

which has a $pH_{pzc}$ of 12.0 [33]. Fe-BC had a low $pH_{pzc}$ (2.1) due to the low $pH_{pzc}$ (2.0) of $Fe_3O_4$ [34]. In the polluted water with a pH of around 7.0, the Mg-BC particles possess the most positive surface charges, while the Fe-BC possess negative surface charges and repulse phosphate.

### 3.1.2. Textural features

Pretreatment of the biomass with the five metal chlorides increased the specific surface area ($S_{BET}$) and total pore volume ($V_{pore}$) of the biochar by 0.5–23.7 and 0.2–2.5 times, respectively, while reducing the average pore diameter ($D_{pore}$) of the biochar from 30.53 to 4.38–26.61 nm (table 2). $MgCl_2$ exhibited the strongest effect on increasing the $S_{BET}$ and $V_{pore}$ of the biochar, followed by $LaCl_3$. The effects of $AlCl_3$, $CaCl_2$ and $FeCl_3$ were much weaker. In addition, Mg-BC had the largest proportions of the specific surface area and pore volume of micropores ($S_{micro}$ and $V_{micro}$) in $S_{BET}$ and $V_{pore}$, which were 72.1% and 34.0%, respectively, indicating that Mg-BC contained many micropores. La-BC and Al-BC also contained a certain amount of micropores, while there were barely any micropores in Fe-BC, Ca-BC and BC. Pretreatment of the biomass with metal chlorides could improve the biochar porosity through two mechanisms. Firstly, the metal chlorides acted as the activation agents during the pyrolysis of the biomass and new micro- (less than 2 nm), meso- (2–50 nm) or macropores (greater than 50 nm) were formed on the walls of the biomass [26]. Secondly, the metal oxides transformed from the metal chlorides and loaded on the biochar matrices also contained many pores. The Mg oxide particles contained mainly micropores [35], while the Fe and Ca oxides contained mainly mesopores [14,31]. Therefore, the different catalysation effects of the different metal chlorides on the biomass pyrolysis and the different porous structures of the formed metal oxides particles contributed together to the different textural features of the produced biochars.

### 3.1.3. Surface functional groups

The FTIR spectra of the biochars are shown in figure 1. The peaks at 3735–3017 cm⁻¹ were assigned to the O–H stretching vibration of hydrogen-bonded hydroxyl groups (−OH) or adsorbed water [26]. The peaks at 2965–2916 and 874–669 cm⁻¹ corresponded to the aromatic C–H vibration [36,37]. The peaks at 1630–1378 cm⁻¹ corresponded to the aromatic C=C and C=O stretching vibrations [37]. Those at 512–422 cm⁻¹ were assigned to the bonds of the metals (Al, Fe, La and Mg) with oxygen [38]. All these functional groups were present on the surface of each biochar except that the metal–oxygen bond was absent on the BC surface. The metal (Al, Fe, La and Mg)–oxygen bonds indicated the incorporation of metals in the biochar matrices. The loading of Ca on the Ca-BC was confirmed by the presence of $Ca(OH)_2$, the peaks of which overlapped the peaks of −OH centred at 3404 cm⁻¹ [39]. CaO was formed during the pyrolysis of the $CaCl_2$-treated biomass and transformed to $Ca(OH)_2$ due to adventitiously chemisorbed moisture that occurred with handling prior to FTIR analysis [11,39]. The peak intensity of the −OH group was much higher in the Ca-BC and Fe-BC than in the other biochars, indicating the larger number of −OH groups in these biochars.

### 3.1.4. Surface crystal composition and morphology

The XRD spectra (figure 2a) showed that AlOOH, $Fe_3O_4$, MgO, and LaOCl and $La_2O_3$ crystals were loaded on the Al-BC, Fe-BC, Mg-BC and La-BC, respectively, while KCl crystals were present on the BC. $CaCO_3$ and CaO crystals were loaded on the Ca-BC. The low intensity of the diffraction peaks of AlOOH crystals on the Al-BC was attributed to the highly dispersed distribution of AlOOH crystals on the biochar surface as well as the low crystallinity and extremely small size of

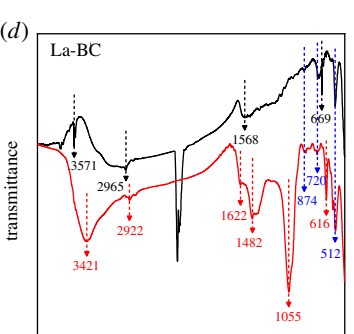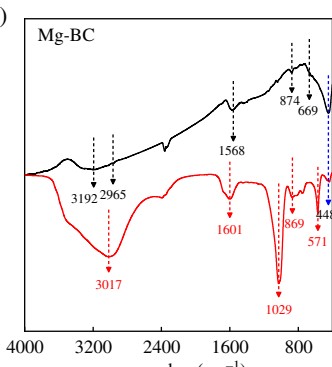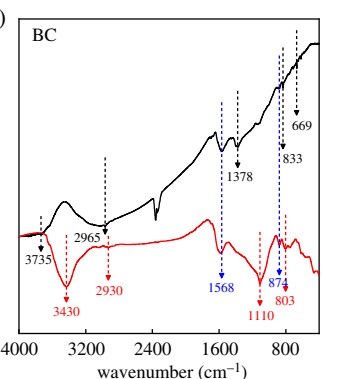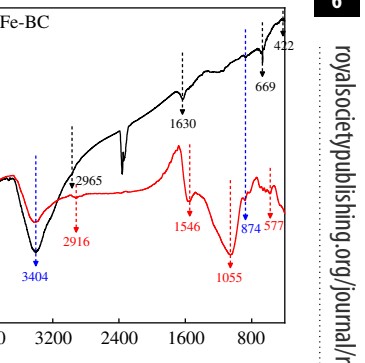

**Figure 1.** FTIR spectra of the biochars before and after phosphate adsorption.

the AlOOH crystallites [12], which was confirmed by the surface morphology of the Al-BC (electronic supplementary material, figure S1). The metal oxides on the five modified biochars were likely formed through a sequence of dechlorination and dehydration reactions such as $MgCl_2 \rightarrow Mg(OH)_2 \rightarrow MgO$ and $FeCl_3 \rightarrow Fe(OH)_3 \rightarrow FeO(OH) \rightarrow Fe_3O_4$ with gas forms of HCl and $H_2O$ produced and released [7].

Surface morphologies of the six biochars were significantly different (electronic supplementary material, figure S1). The BC surface was smooth and maintained the intrinsic nature of a part of the biomass. The surfaces of the modified biochars were rough, and the metal oxides generated during the pyrolysis not only covered the surfaces but also entered the inner pores of the biochars. Specifically, the inner and outer surfaces of Mg-BC were covered with a large number of crystals in the shape of needles and thin sheets, the typical morphology of MgO crystals [33]. Numerous short rod-like crystals, which were $La_2O_3$ or LaOCl from the XRD analysis, distributed on the surface and inside the pores of the La-BC. Fine AlOOH granules dispersed on the surface of the Al-BC. The surfaces and pores of the Ca-BC and Fe-BC were coated or filled with lumpy matters, which were agglomerated CaO and $Fe_3O_4$, respectively, as shown by the XRD analysis. The surface morphologies of the biochars were consistent with their textural properties, i.e. the Mg-BC and La-BC with a large number of porous MgO or $La_2O_3$/LaOCl particles on the surface had large specific surface areas and many micropores, while the Al-BC, Ca-BC, Fe-BC and BC with a small number of particles or many agglomerated particles on the surface had small specific surface areas.

## 3.2. Phosphate adsorption kinetics

Whereas the BC kept releasing phosphate to the solution, the five modified biochars showed different patterns of phosphate adsorption kinetics (figure 3). In the preliminary experiments where 0.2 g of the six biochars was placed in 80 ml deionized water, respectively, and shaken at 25°C and 200 r $min^{-1}$ for 24 h, release of phosphate from the BC to the deionized water was also observed since P in the reed was retained in the BC. However, negligible phosphates were released from the five modified biochars to the deionized water. Labile phosphates in the reed transformed to less soluble forms during the pyrolysis of the metal chlorides and impregnated reed by binding with the metal ions [40]. In order to

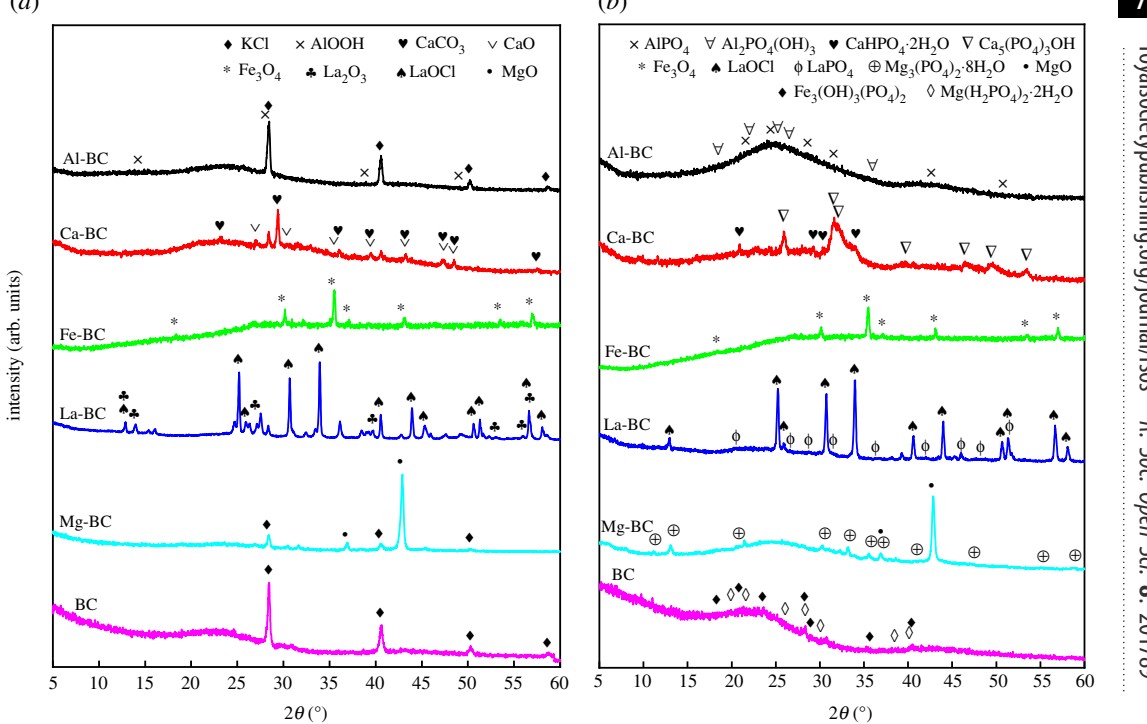

**Figure 2.** XRD spectra of the biochars (a) before and (b) after phosphate adsorption.

study the phosphate adsorption mechanisms of the modified biochars, the experimental data were fitted with several empirical kinetic models [9]

$$Q_t = Q_e(1 - \exp(-k_1 t)) \quad \text{Pseudo-first-order}, \tag{3.1}$$

$$Q_t = \frac{k_2 Q_e^2 t}{(1 + k_2 Q_e t)} \quad \text{Pseudo-second-order}, \tag{3.2}$$

$$Q_t = Q_e - [Q_e^{1-n} + (n-1)k_n t]^{1/(1-n)} \quad \text{Ritchie } n\text{th-order}, \tag{3.3}$$

$$Q_t = \frac{1}{\beta\Delta}\ln(1 + \alpha\beta t) \quad \text{Elovich} \tag{3.4}$$

and
$$Q_t = k_i\sqrt{t} + C \quad \text{Intra-particle-diffusion}, \tag{3.5}$$

where $Q_t$ (mg g$^{-1}$) and $Q_e$ (mg g$^{-1}$) are the amount of $PO_4^{3-} - P$ adsorbed at time $t$ and equilibrium, respectively, $k_1$ (h$^{-1}$), $k_2$ (g (mg h)$^{-1}$), $k_n$ (g$^{n-1}$ (mg$^{n-1}$h)$^{-1}$) and $k_i$ (mg (g h$^{0.5}$)$^{-1}$) are the first-, second- and $n$th-order kinetic and the intra-particle diffusion rate constants, respectively, $\alpha$ (mg (g h)$^{-1}$) is the initial adsorption rate, $\beta$ (g mg$^{-1}$) is the desorption constant and $C$ is a constant. The correlation coefficient ($R^2$) and residue $\chi^2$ error were employed to compare the applicability of the models and determine the best-fitting one [13].

La-BC had the largest phosphate adsorption rate, and the phosphate adsorption amount reached the quasi-equilibrium within 1.0 h. The pseudo-second-order model fitted best to the phosphate adsorption kinetic data of La-BC with the highest $R^2$ (0.817), smallest $\chi^2$ ($1.25 \times 10^{-4}$) and the simulated equilibrium adsorption amount of 9.90 mg g$^{-1}$ being close to the measured value of 9.87 mg g$^{-1}$ (table 3), suggesting that phosphate adsorption on the La-BC could be controlled by chemisorption [38].

Al-BC had the smallest phosphate adsorption rate among the five modified biochars. The amount of phosphate adsorbed by the Al-BC kept increasing with a decreasing rate without reaching an equilibrium within 72 h. The phosphate adsorption kinetic data of Al-BC were best fitted with the Elovich model, indicating that phosphate was adsorbed on highly heterogeneous surfaces via chemisorption [41]. The phosphate adsorption kinetic data of Al-BC were also well fitted with the intra-particle diffusion model ($R^2 = 0.988$) and the intercept of the fitting line ($C = 1.40$) was not much larger than 0 (figure 4; electronic supplementary material, table S1), suggesting that the small phosphate adsorption rate of Al-BC was mainly due to the slow diffusion of phosphate ions inside

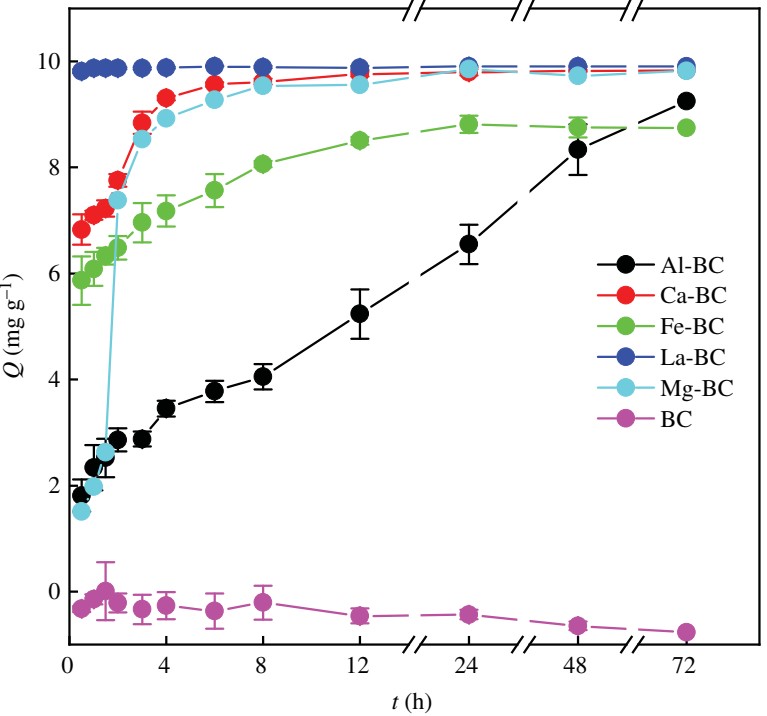

**Figure 3.** Phosphate adsorption kinetics of the biochars (initial phosphate concentration 20 mg l$^{-1}$ PO$_4^{3-}$ — P, biochar concentration 2 g l$^{-1}$, initial solution pH 7.0, temperature 25℃, shaking speed 150 r min$^{-1}$, reaction time 72 h).

**Table 3.** Adsorption kinetic parameters for phosphate adsorption on the biochars.

| biochar | | Al-BC | Ca-BC | Fe-BC | La-BC | Mg-BC |
|---|---|---|---|---|---|---|
| $Q_{e\_exp}$[a] (mg g$^{-1}$) | | —[b] | 9.76 | 8.82 | 9.87 | 9.86 |
| pseudo-first-order model | $Q_e$ (mg g$^{-1}$) | 8.17 | 9.35 | 7.88 | 9.89 | 9.94 |
| | $k_1$ (h$^{-1}$) | 0.11 | 1.63 | 1.75 | 9.81 | 0.44 |
| | $R^2$ | 0.808 | 0.503 | 0.333 | 0.617 | 0.890 |
| | $\chi^2$ | 1.13 | 0.72 | 0.83 | $2.62 \times 10^{-4}$ | 1.20 |
| pseudo-second-order model | $Q_e$ (mg g$^{-1}$) | 9.25 | 9.84 | 8.42 | 9.90 | 10.92 |
| | $k_2$ (g (mg h)$^{-1}$) | 0.02 | 0.31 | 0.32 | 24.00 | 0.05 |
| | $R^2$ | 0.885 | 0.843 | 0.757 | 0.817 | 0.818 |
| | $\chi^2$ | 0.68 | 0.23 | 0.30 | $1.25 \times 10^{-4}$ | 2.00 |
| Ritchie $n$th-order model | $Q_e$ (mg g$^{-1}$) | —[c] | 10.46 | 12.37 | 9.90 | 9.96 |
| | $k_n$ (g$^{n-1}$ (mg$^{n-1}$ h)$^{-1}$) | | 0.05 | $5.82 \times 10^{-7}$ | 33.09 | 0.45 |
| | $n$ | | 2.98 | 7.82 | 2.20 | 0.97 |
| | $R^2$ | | 0.868 | 0.931 | 0.802 | 0.880 |
| | $\chi^2$ | | 0.19 | 0.085 | $1.36 \times 10^{-4}$ | 1.31 |
| Elovich model | $\alpha$ (mg (g h)$^{-1}$) | 2.72 | 41957 | 5442 | —[c] | 22.34 |
| | $\beta$ (g mg$^{-1}$) | 0.53 | 1.45 | 1.43 | | 0.58 |
| | $R^2$ | 0.957 | 0.760 | 0.924 | | 0.627 |
| | $\chi^2$ | 0.25 | 0.35 | 0.095 | | 4.09 |

[a]Equilibrium phosphate adsorption amount measured in the experiment.

[b]Equilibrium was not reached within the experimental time.

[c]Fit did not converge.

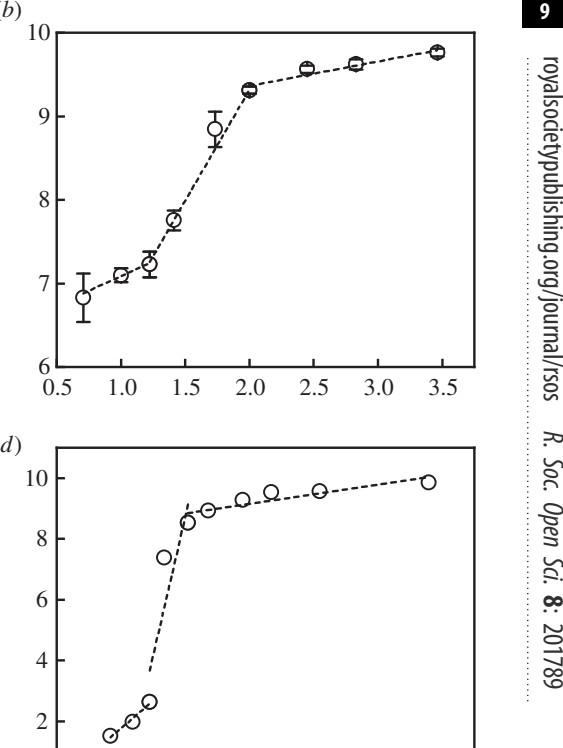

**Figure 4.** Intra-particle diffusion model fitting curves of (a) Al-BC, (b) Ca-BC, (c) Fe-BC and (d) Mg-BC.

the Al-BC particle before they could contact with the adsorption sites [12]. The specific surface area of Al-BC was the smallest among the five modified biochars and the total pore volume of Al-BC was close to the smallest value (table 2), indicating that the porosity of Al-BC was not well developed. Additionally, the main adsorption sites of Al-BC, the AlOOH particles, distributed highly dispersedly on the surface. These two factors made the adsorption sites difficult to access and resulted in the small phosphate adsorption rate.

Ca-BC and Fe-BC showed much larger phosphate adsorption rates in the initial 0.5 h than the later time. The amounts of phosphate adsorbed by the Ca-BC and Fe-BC in the first 0.5 h accounted for 69.7% and 66.5%, respectively, of their equilibrium adsorption amounts reached within 12 h and 24 h, respectively. The large initial adsorption rates were attributed to the availability of a large number of easily accessible vacant adsorption sites on the biochar surfaces. The Ritchie $n$th-order model fitted best to the kinetic data of Ca-BC and Fe-BC, indicating that phosphate adsorption on the two biochars was controlled by multiple interactions [42]. The phosphate adsorption data of Fe-BC during 0.5-24 h were also well fitted with the intra-particle diffusion model ($R^2 = 0.926$, $C = 5.50$). Therefore, after the fast adsorption phase in the first 0.5 h, the phosphate adsorption rate of Fe-BC was restricted by the diffusion of phosphate ions outside and inside the Fe-BC particle. Fitting the phosphate adsorption data of Ca-BC with the intra-particle diffusion model showed multi-linearity, implying that two or more steps occurred during the process [32]. The phosphate adsorption rate of Ca-BC was controlled by phosphate diffusion outside and inside the particle during 0.5–4 h ($R^2 = 0.969$–0.977, $C = 3.85$–6.30).

The phosphate adsorption rate of Mg-BC increased sharply during 1.5–2.0 h after a slower initial phase and decreased afterwards until the equilibrium was reached within 24 h. The pseudo-first-order model fitted best to the phosphate adsorption kinetic data of Mg-BC, indicating that phosphate adsorption on the Mg-BC could be controlled by physical adsorption [26], which might be related to the large specific surface area and high $pH_{pzc}$ of Mg-BC. The small phosphate adsorption rate of Mg-BC in the initial 0.5–1.5 h was due to the small intra-particle diffusion rate of phosphate, as indicated by the high correlation coefficient ($R^2 = 0.944$) and small value of intercept ($C = -0.05$) of the fitted intra-particle diffusion model [26]. Afterwards, intra-particle diffusion was not rate-limiting

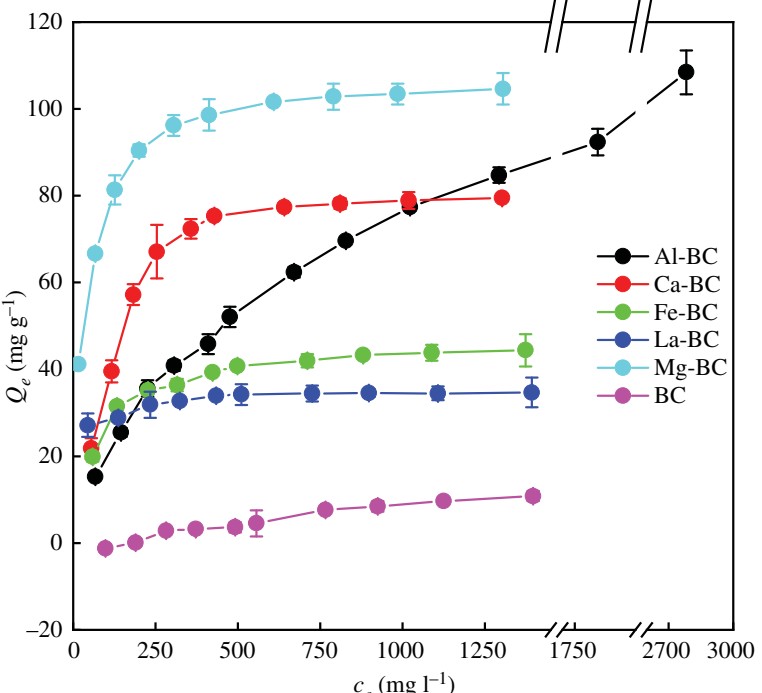

**Figure 5.** Phosphate adsorption isotherm of the biochars (initial phosphate concentration 100–3000 mg $l^{-1}$ $PO_4^{3-}$ − P for Al-BC and 100–1500 mg $l^{-1}$ for the other five biochars, biochar concentration 2 g $l^{-1}$, initial solution pH 7.0, temperature 25°C, shaking speed 150 r $min^{-1}$, reaction time 168 h for Al-BC and 24 h for the other biochars).

($R^2 = 0.567$–0.741). Gong et al. [43] also observed a sharp increase in the phosphate adsorption rate 4 h after the biochar pyrolysed from $MgCl_2$ pretreated *P. australis* was added to the solution. The sharp increase in the phosphate adsorption rate could be due to the surface precipitation of phosphate with MgO on the biochar [9].

Xu et al. [25] prepared four biochars pyrolysed from $AlCl_3$, $CaCl_2$, $FeCl_3$ and $MgCl_2$, respectively, pretreated wood waste and compared the phosphate removal efficiencies of them from diluted urine containing 31.8 mg $l^{-1}$ $PO_4^{3-}$ − P. The $MgCl_2$ modified one exhibited the highest phosphate removal efficiency (97.2%) after the reaction time of 24 h, followed by the $AlCl_3$ (14.6%), $FeCl_3$ (12.7%) and $CaCl_2$ (6.3%) modified ones. The $PO_4^{3-}$ − P concentration in the kinetic experiments of the present study (20 mg $l^{-1}$) did not deviate largely from that in Xu et al. [25]. However, much higher phosphate removal efficiencies were achieved for Al-BC, Ca-BC and Fe-BC after 24 h, which were 66.0%, 98.4% and 88.9%, respectively. Mg-BC exhibited higher phosphate removal efficiency (99.4%) than Al-BC, Ca-BC and Fe-BC, similar to the results of Xu et al. [25]. However, the removal efficiency of Ca-BC was only slightly lower than Mg-BC, and Al-BC had the lowest phosphate removal efficiency due to the slow adsorption rate. The higher phosphate removal efficiency in the present study could be partly due to the absence of interfering substances in the synthetic solution, while the diluted urine contained many other organic and inorganic substances besides phosphate which could compete for or block the adsorption sites [17]. In addition, the different types of biomass feedstock [11], amounts of metals added to the biomass [15] and pyrolysis conditions result in different characteristics of produced biochars, which also contribute to the different results from the two studies.

## 3.3. Phosphate adsorption isotherms

Phosphate adsorption isotherms of the six biochars are shown in figure 5. As the initial $PO_4^{3-}$ − P concentration increased from 100 to 1500 mg $l^{-1}$, the phosphate adsorption amount of BC rose from −1.2 to 10.8 mg $g^{-1}$. The phosphate adsorption amount of Ca-BC, Fe-BC, La-BC and Mg-BC increased until reaching the maximum value (the maximum adsorption capacity) which were 79.5, 44.4, 34.7 and 104.6 mg $g^{-1}$, respectively. The phosphate adsorption amount of Al-BC kept increasing without reaching a plateau in the initial $PO_4^{3-}$ − P concentration range of 100–3000 mg $l^{-1}$.

**Table 4.** Adsorption isotherm parameters for phosphate adsorption on the biochars.

| biochar | | Al-BC | Ca-BC | Fe-BC | La-BC | Mg-BC | BC |
|---|---|---|---|---|---|---|---|
| $Q_{m-exp}$[a] | | —[b] | 79.5 | 44.4 | 34.7 | 104.6 | —[b] |
| Langmuir model | $Q_m$ | 122.2 | 91.6 | 46.6 | 34.7 | 106.3 | —[c] |
| | $k_L$ ($\times 10^{-3}$) | 1.64 | 8.13 | 13.49 | 67.66 | 30.21 | |
| | $R^2$ | 0.990 | 0.947 | 0.988 | 0.838 | 0.979 | |
| | $\chi^2$ | 4.97 | 20.54 | 0.68 | 1.15 | 8.94 | |
| Freundlich model | $1/n$ | 0.54 | 0.27 | 0.20 | 0.07 | 0.17 | 1.02 |
| | $k_F$ | 1.86 | 13.27 | 11.52 | 20.84 | 34.12 | 0.01 |
| | $R^2$ | 0.994 | 0.764 | 0.875 | 0.885 | 0.869 | 0.926 |
| | $\chi^2$ | 2.83 | 91.17 | 7.06 | 0.81 | 55.40 | 1.2 |
| Langmuir–Freundlich model | $Q_m$ | 219.9 | 81.5 | 46.6 | 38.9 | 112.5 | 13.5 |
| | $1/n$ | 0.72 | 1.68 | 1.01 | 0.43 | 0.77 | 2.07 |
| | $k$ ($\times 10^{-3}$) | 3.77 | 0.38 | 13.19 | 434.07 | 63.94 | 0.001 |
| | $R^2$ | 0.998 | 0.992 | 0.986 | 0.920 | 0.993 | 0.957 |
| | $\chi^2$ | 1.24 | 3.21 | 0.77 | 0.57 | 2.97 | 0.70 |

[a]Maximum adsorption capacity measured in the experiment.
[b]Maximum adsorption capacity was not reached in the experimental range of initial phosphate concentration.
[c]Fit did not converge.

The isotherm experimental data of the biochars were fitted with the Langmuir, Freundlich and Langmuir–Freundlich models [9]

$$Q_e = \frac{k_L Q_m c_e}{(1 + k_L c_e)} \quad \text{Langmuir,} \tag{3.6}$$

$$Q_e = k_F c_e^{1/n} \quad \text{Freundlich} \tag{3.7}$$

and

$$Q_e = \frac{k Q_m c_e^{1/n}}{(1 + k c_e^{1/n})} \quad \text{Langmuir} - \text{Freundlich,} \tag{3.8}$$

where $Q_m$ (mg g$^{-1}$) is the maximum PO$_4^{3-}$ – P adsorption capacity, $c_e$ (mg l$^{-1}$) is the equilibrium PO$_4^{3-}$ – P concentration in the solution, $k_L$ (l mg$^{-1}$), $k_F$ (mg l$^{1/n}$ (g mg$^{1/n}$)$^{-1}$) and $k$ (l$^{1/n}$ (mg$^{-1/n}$) are the Langmuir, Freundlich and Langmuir–Freundlich constants, respectively, and $1/n$ is a parameter indicating the reaction strength between the adsorbate and adsorbent.

The model fitting results (table 4) show that the phosphate adsorption isotherm of Fe-BC was best fitted with the Langmuir model and the isotherms of the other five biochars were best fitted with the Langmuir–Freundlich model with the highest $R^2$ and smallest $\chi^2$. The maximum phosphate adsorption capacities of Ca-BC (81.5 mg g$^{-1}$), Fe-BC (46.6 mg g$^{-1}$), La-BC (38.9 mg g$^{-1}$) and Mg-BC (112.5 mg g$^{-1}$) predicted from the Langmuir–Freundlich or Langmuir model were close to the measured values. These results suggest that the adsorption of phosphate by the Fe-BC occurred on a uniform surface with constant energy [22], while the adsorption of phosphate by the other five biochars occurred on heterogeneous surfaces and multiple processes were involved [12,21]. The maximum phosphate adsorption capacities of Al-BC and BC predicted from the Langmuir–Freundlich model were 219.9 and 13.5 mg g$^{-1}$, respectively. Based on the model fitting results, the maximum phosphate adsorption capacities of the biochars followed the order of Al-BC > Mg-BC > Ca-BC > Fe-BC > La-BC > BC.

Smaller $1/n$ values of the Langmuir–Freundlich model indicate stronger interactions between the adsorbate and adsorbent [21]. $1/n$ was the smallest for La-BC, indicating that La-BC had the strongest interaction with phosphate ions, which might be attributed to the high phosphate affinity of La [38]. BC with no active metal oxide particles on the surface had the weakest interaction with phosphate ions as indicated by the largest $1/n$ value.

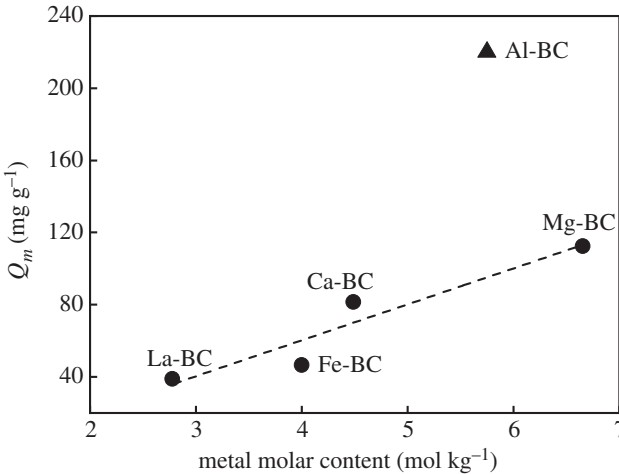

**Figure 6.** Relationship between metal molar contents and maximum phosphate adsorption capacities of the biochars.

## 3.4. Factors determining the phosphate adsorption capacities and rates of the biochars

The maximum adsorption capacity of an adsorbent depends on the total number of available active adsorption sites [44] which might be related to the amount of metal oxides, surface functional groups and specific surface areas for the metal oxides loaded biochars. Statistical analysis showed that the maximum phosphate adsorption capacities ($Q_m$) of Ca-BC, Fe-BC, La-BC and Mg-BC were linearly correlated with the molar contents of the corresponding metals in the biochars ($R^2 = 0.905$, $p = 0.049$, figure 6), while $Q_m$ values of the five modified biochars were minimally correlated with the O/C molar ratios ($R^2 = 0.419$, $p = 0.238$) and the specific surface areas ($R^2 = 0.096$, $p = 0.611$). These results implied that the metal oxides loaded on the biochars were the main adsorption sites for phosphate, in agreement with previous studies [24]. Moreover, the metal molar content determined the phosphate adsorption capacities of the Ca-BC, Fe-BC, La-BC and Mg-BC irrespective of the type of metal oxides. Nevertheless, Al-BC showed a higher phosphate adsorption capacity than the other modified biochars at the identical metal molar content. This could be explained by the phosphate adsorption mechanisms of the biochars.

As shown in figure 2b, new crystals in the forms of $AlPO_4$ and augelite ($Al_2PO_4(OH)_3$), hydroxylapatite ($Ca_5(PO_4)_3(OH)$) and brushite ($CaHPO_4 \cdot 2H_2O$), $LaPO_4$ and $Mg_3(PO_4)_2 \cdot 8H_2O$ were identified on the Al-BC, Ca-BC, La-BC and Mg-BC, respectively, after phosphate adsorption. During the adsorption process, phosphate reacted with the $Al^{3+}$, $Ca^{2+}$, $La^{3+}$ and $Mg^{2+}$ ions released from the metal oxides to form these crystals which nucleated and grew on the biochar surface due to the low surface energy [9,45]. The phosphate/metal molar ratios of the formed precipitates were between 0.6 and 1.0 except augelite ($Al_2PO_4(OH)_3$) with a phosphate/metal molar ratio of 2.0. Precipitation was not an important mechanism of phosphate adsorption by the Fe-BC, since new forms of crystals were not detected on the Fe-BC after phosphate adsorption. The FTIR analysis of the biochars after phosphate adsorption showed that new peaks in the ranges of 1157–1029 cm$^{-1}$ and 616–563 cm$^{-1}$, corresponding to the P–O stretching and O–P–O bending vibrations, respectively [15,36], appeared on the six biochars, conforming the adsorption of phosphate on the biochars. Compared with the peak intensities before adsorption, the Al-O, La-O and Mg-O peaks of the Al-BC, La-BC and Mg-BC, respectively, weakened after adsorption while their −OH peaks strengthened, which could be explained by the transformation of some Al, La and Mg oxides into hydroxyl compounds in the solution through hydrolysis reactions. For the Ca-BC and Fe-BC, the −OH peaks of the two biochars and the Fe–O peak of Fe-BC weakened, indicating the involvement of the metal–O and −OH groups in the adsorption process. When interacting with the phosphate solution, the metal–O and −OH groups of the biochars bound or exchanged with phosphate, forming mono-, bi- and tri-nuclear inner-sphere surface complexes with the phosphate/metal molar ratio of close to 1.0 [17,26,33]. The good fit of the phosphate adsorption isotherm of Fe-BC with the Langmuir model suggested that phosphate was adsorbed on a homogenous surface, i.e. through a single mechanism. Therefore, phosphate was probably adsorbed on the Fe-BC mainly by formation of complexes. For Al-BC, Ca-BC, La-BC and Mg-BC, surface precipitation and complexes formation were the main phosphate adsorption mechanisms, as indicated by the good fit of their phosphate adsorption isotherms with the

Langmuir–Freundlich model. The close phosphate/metal molar ratios of the precipitates and complexes formed between phosphate and the Ca-BC, Fe-BC, La-BC and Mg-BC (0.6–1.0) led to the result that $Q_m$ of the four biochars was linearly correlated with the metal molar content irrespective of the type of metal oxide. A precipitate with a higher phosphate/metal molar ratio was formed between phosphate and Al-BC, resulting in the higher $Q_m$ of Al-BC.

Previous studies have reported that the phosphate adsorption capacities of biochars loaded with the same type of metal oxides improved with increasing metal content of the biochar or amount of metals added to the biomass for modification [15]. The linear correlation between the maximum phosphate adsorption capacities of the biochars loaded with Ca, Fe, La and Mg oxides, respectively, and the molar contents of these metals in the biochars found in this study indicates that the Ca, Fe, La and Mg chlorides are substitutional to be used to prepare the metal oxide-loaded biochars regarding the phosphate adsorption capacity. However, the phosphate adsorption kinetics of the biochars loaded with different metal oxides varied significantly.

The intra-particle model fitting results showed that the phosphate diffusion rate outside and/or inside the biochar particle controlled the phosphate adsorption rates of Al-BC during 0.5–72 h, Fe-BC during 0.5–24 h, Ca-BC during 0.5–4 h and Mg-BC during 0.5–1.5 h. The phosphate adsorption rates of the four biochars during 0.5–1.5 h had a weak correlation with the total pore volumes ($V_{pore}$) of the biochars ($R^2 = 0.649$, $p = 0.195$). This was due to the fact that $V_{pore}$ was related to the phosphate diffusion rate inside the particle, while the phosphate adsorption rates of Fe-BC and Ca-BC were controlled by a combination of phosphate diffusion rates outside and inside the particle. Besides the porosity, the pore size distribution and arrangement or structure of the pores also affects the diffusion of an adsorbent ion through the adsorbent [44,45]. More research is needed to reveal the factors determining the phosphate adsorption rates of the biochars loaded with different metal oxides.

# 4. Conclusion

The biochars loaded with different metal oxides differ significantly in the chemical composition, textural features, and phosphate adsorption kinetics and isotherms. The maximum phosphate adsorption capacities of Ca-BC, Fe-BC, La-BC and Mg-BC depend on the molar content of the metals in the biochars. The Al-BC has the highest phosphate adsorption capacity but the smallest phosphate adsorption rate which is due to the slow intra-particle diffusion of phosphate attributed to the undeveloped porosity and highly dispersed distribution of AlOOH crystals on the Al-BC surface. Mg-BC is suggested for phosphate removal from water considering both phosphate adsorption rate and capacity. Al-BC is suitable to be used when long contact time is allowed, e.g. as a capping material to immobilize phosphate in the sediment of eutrophic lakes. In further studies, effects of various competing constituents in the real polluted water on the phosphate adsorption characteristics of the metal oxide-loaded biochars and stability of the biochars in the real polluted water need to be clarified before applying them in practice.

Data accessibility. Data available from the Dryad Digital Repository: https://doi.org/10.5061/dryad.3xsj3txf4 [46].

Authors' contributions. P.W. designed the study, analysed the data and drafted the manuscript. M.Z. ran the experiments, analysed the data and helped draft the manuscript. G.C. helped draft the manuscript. Z.C. helped design the study and revised the manuscript. S.W. conceived of the study, coordinated the study and revised the manuscript. All authors gave final approval for publication and agree to be held accountable for the work performed therein.

Competing interests. We declare we have no competing interests.

Funding. This work was supported by the Major Science and Technology Program for Water Pollution Control and Treatment (grant no. 2017ZX07206-003) and the Research Initiation Fund for Young Teachers of Beijing Technology and Business University (grant no. PXM2018_014213_000033).

Acknowledgements. We thank Dr Joseph Flora from University of South Carolina for advices and discussions in the statistical analysis.

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
