## [Peer Review File · Royal Society Open Science]

Review History

RSOS-201789.R0 (Original submission)

Review form: Reviewer 1

Is the manuscript scientifically sound in its present form?

Yes

Are the interpretations and conclusions justified by the results?

Yes

Is the language acceptable?

Yes

Do you have any ethical concerns with this paper?

No

Have you any concerns about statistical analyses in this paper?

No

Recommendation?

Major revision is needed (please make suggestions in comments)

Comments to the Author(s)

I have carefully read the manuscript RSOS-201789 entitled "A comparative study on phosphate removal from water using phragmites australis biochars loaded with different metal oxides" and find the research work relevant, fitting the scope of the journal and likely to be of interest to a broad readership. The research goals, methodology and results/discussion are well structured, the research hypotheses and scientific interpretations are very well justified, and the research work has sufficient novelty. In my opinion, however, the manuscript needs some major revisions before being published in the Royal Society Open Science. Authors may consider the following recommendations to improve the quality of the manuscript:

1) Introduction, Page 3, Lines 1-2: A comprehensive study demonstrating the suitability of adsorption to remove and even recover phosphorus from low concentrated secondary effluent wastewater, including a verification of the long-term reusability of the adsorbents, is recommended:

"Polishing of secondary wastewater effluents through elimination and recovery of dissolved phosphorus with reusable magnetic microsorbents". Proceedings of the Water Environment Federation (WEF): Nutrient Symposium 2017 (3), ISSN 1938-6478, 169–181, DOI: <https://doi.org/10.2175/193864717821494169>

2) Introduction, Page 3, Lines 9-10: The following very systematic screening study representing the high phosphorus adsorption and desorption efficiency of various metal (oxy)hydroxides can be added here:

"Influence of cation building blocks of metal hydroxide precipitates on their adsorption and desorption capacity for phosphate in wastewater - A screening study". Colloids and Surfaces A: Physicochem. and Eng. Aspects, 488, 145–153. DOI: <https://doi.org/10.1016/j.colsurfa.2015.10.017>

3) Page 3, Line 48: Note down also the respective concentration of the biochar in parenthesis: i.e. 1 g biochar in 500 mL KH₂PO₄ solution corresponds to 2 g/L biochar. The same applies to Page 4, Line 2: 0.08 g biochar in 40 mL KH₂PO₄ solution corresponds to 2 g/L biochar.

4) Page 4, Lines 27-28: Authors state that the fitting of the kinetic and isotherm models to the experimental data was performed with Origin 2018 using the non-linear fitting methods, as plotted in Figure 3 and Figure 5. Why didn't authors simply plot the smooth fitting curves based on the actual modeling parameters calculated from the respective model equations (Tables 3 and 4)?

5) Page 5, Lines 5-6: Authors measured that most of the metal-doped biochars had pHPZC around the neutral range, except for Fe-BC with pHPZC=2.1 and Mg-BC with pHPZC=11.3. They explained this with the high pHPZC~12 of MgO and likely due to the low pHPZC of Fe₃O₄ (magnetite). In support of this statement, I can recommend the paper by Mandel et al. where the researchers measured the pHPZC of various Mg-based oxyhydroxides which all had pHPZC~12, and they also measured the pHPZC of magnetite (Fe₃O₄) nanoparticles which showed pHPZC~2:

Mandel, K. et al. (2013). "Layered double hydroxide ion exchangers on superparamagnetic microparticles for recovery of phosphate from waste water". Journal of Materials Chemistry A 1 (5), 1840-1848, DOI: <https://doi.org/10.1039/C2TA00571A>

6) Table 1: There is a typo in the column named "Mg-C". Please, replace with "Mg-BC".

- 7) Figure 3: It would help the reader if authors include in the figure caption a remark, listing the reaction parameters of the kinetics experiment, i.e. initial P-concentration: 20 mg/L PO₄-P; biochar concentration 2 g/L, pH 7, 25°C, etc. The same refers to Figure 5 (isotherms) where authors can list the phosphate and biochar concentration range, contact time, pH, temperature, etc.
- 8) Authors performed the study only in a synthetic KH₂PO₄ solution. Nevertheless, they need to keep in mind that the biochar performance in a real wastewater or water matrix may lead to completely different results. The competing effect of other constituents (dissolved ions, organics, suspended matter, etc.) present in real water / wastewater matrices must be taken into account in future studies.
- 9) Have authors checked the chemical and mechanical stability of the developed metal-incorporated biochars? They need to prove that none of the metals or other precursors from the Me-BC are leached into the treated effluent, especially if they carry a phosphate ion with them, which would essentially pollute the effluent instead of purifying it.
- 10) How do authors imagine the practical engineering application of the metal-loaded biochars, i.e. how would the biochars be harvested / separated from the treated wastewater? The efficiency of the biochar separation will directly affect the effluent quality. If a simple sedimentation is proposed, probably there will be a need to dose flocculants to assure an efficient sedimentation process, but the flocculants may change the properties of the P-rich biochars and make them unsuitable for field applications, e.g. as soil amendment. Generally, in the future the authors need to investigate the plant availability and fertilizer value of the proposed P-loaded biochars. Often times the crystalline metal-phosphate precipitates, e.g. AlPO₄, are not suitable as fertilizers and have poor plant availability, which might defeat the original concept of this study to utilize the P-loaded biochars directly on the field!

Review form: Reviewer 2

Is the manuscript scientifically sound in its present form?

No

Are the interpretations and conclusions justified by the results?

Yes

Is the language acceptable?

Yes

Do you have any ethical concerns with this paper?

No

Have you any concerns about statistical analyses in this paper?

No

Recommendation?

Accept with minor revision (please list in comments)

Comments to the Author(s)

This work actually lots of effective experimental data. Publication of these data in public open access journal media would have good contribution to the related research communities. From

the latter positive point, I basically recommend publication of this manuscript in Royal Society Open Science. Please see below necessary revisions.

1) Adsorption results were well reported, but mechanisms are not fully clear. The latter matters would be revealed by appropriate spectral analyses. IR measurement of these bio-derived carbon materials (with metal oxides) with phosphates had better be measured. From the peak shifts, interaction between the materials and phosphates can be discussed.

2) References can be more generalized and updated. For example, biomass-derived carbon materials are hot subjects and related papers can be cited more (for example, see below).
<https://www.journal.csj.jp/doi/10.1246/bcsj.20200055>
<https://www.sciencedirect.com/science/article/abs/pii/S030438941931194X?via%3Dihub>

3) Please provide clear scale bars to images in Figure S1.

Review form: Reviewer 3

Is the manuscript scientifically sound in its present form?

No

Are the interpretations and conclusions justified by the results?

No

Is the language acceptable?

Yes

Do you have any ethical concerns with this paper?

No

Have you any concerns about statistical analyses in this paper?

No

Recommendation?

Major revision is needed (please make suggestions in comments)

Comments to the Author(s)

In the manuscript entitled "A comparative study on phosphate removal from water using phragmites australis biochars loaded with different metal oxides", the authors investigated and compared the characteristics and phosphate adsorption kinetics and capacities of the biochars loaded with different metal oxides, and identified possible factors determining the phosphate adsorption capacities and rates of the modified biochars. Overall, most parts of this submission are satisfactory, however, there are some concerns and issues which need be addressed appropriately prior to a possible inclusion in Royal Society Open Science. Detailed comments and suggestions are listed as follows.

On Introduction

1. More better and overall background information for biochars adsorbent should be introduced to the prospective audiences, and the following literature could serve this purpose in some aspects.

Wang S, Characterization and Pb(II) removal potential of corn straw- and municipal sludge-derived biochars. R. Soc. Open. Sci ,2017,4(9),170402.

Wu SH, Role of biochar on composting of organic wastes and remediation of contaminated soils-a review. *Environmental Science and Pollution Research*, 2017, 24(20), 16560-16577.

On Materials and Methods

2. Line 28-30 on Page 3, the first sentence was about straws of *phragmites australis*, the authors should put it on Introduction.
3. Line 50-51 on Page 3, "35 mL samples were taken from each flask at 0.5, 1, 1.5, 2, 3, 4, 6, 8, 12, 24, 48 and 72 h", the authors should clear present if batch experiment was performed in phosphate adsorption kinetic experiments. If not, the changes of solution volume will affect the accuracy of results.
4. pH was measured in phosphate adsorption kinetic experiments and isotherm experiments, but it was not analyzed in the discussion section.
5. The authors should provide data on reusability and regeneration of the adsorbent, and discuss the disposal of the adsorbent after use.

On Conclusion

6. To match the purpose of the research, the authors should show the results of phosphate adsorption rates of the modified biochars on conclusion.
7. When the modified biochars was used "as a capping material to immobilize phosphate in the sediment of eutrophic lakes". One major issue is regarding the metal ion leaching solubility when evaluating its practical application.

Decision letter (RSOS-201789.R0)

This year has been very difficult for everyone, and we want to take the opportunity to thank you for your continued support in 2020.

The Royal Society Open Science editorial office will be closed from the evening of Friday 18 December 2020 until Monday 4 January 2021. We will not be responding during this time. If you have received a deadline within this time period, please contact us as soon as possible to allow us to extend the deadline. If you receive any automated messages during this time asking you to meet a deadline, we offer apologies and invite you to respond after the festive period or during normal working hours.

With our best for a peaceful festive period and New Year, and we look forward to working with you in 2021.

Dear Dr Wang:

Title: A comparative study on phosphate removal from water using *phragmites australis* biochars loaded with different metal oxides

Manuscript ID: RSOS-201789

The editor assigned to your manuscript has now received comments from reviewers. We would like you to revise your paper in accordance with the referee and Subject Editor suggestions which can be found below (not including confidential reports to the Editor). Please note this decision does not guarantee eventual acceptance.

Please submit your revised paper before 31-Jan-2021. Please note that the revision deadline will expire at 00.00am on this date. If we do not hear from you within this time then it will be assumed that the paper has been withdrawn. In exceptional circumstances, extensions may be possible if agreed with the Editorial Office in advance. We do not allow multiple rounds of revision so we urge you to make every effort to fully address all of the comments at this stage. If deemed necessary by the Editors, your manuscript will be sent back to one or more of the original reviewers for assessment. If the original reviewers are not available we may invite new reviewers.

On behalf of the Subject Editor Professor Anthony Stace and the Associate Editor Dr Nadia Martinez Villegas.

RSC Associate Editor:

Comments to the Author:

Thank you for considering RSOS for your manuscript submission. Your manuscript has been carefully examined and, in view of the criticisms of the reviewers, major revision is needed. We look forward to seeing the revised version.

RSC Subject Editor:

Comments to the Author:

(There are no comments.)

Reviewers' Comments to Author:

Reviewer: 1

Comments to the Author(s)

I have carefully read the manuscript RSOS-201789 entitled "A comparative study on phosphate removal from water using phragmites australis biochars loaded with different metal oxides" and find the research work relevant, fitting the scope of the journal and likely to be of interest to a broad readership. The research goals, methodology and results/discussion are well structured, the research hypotheses and scientific interpretations are very well justified, and the research work has sufficient novelty. In my opinion, however, the manuscript needs some major revisions before being published in the Royal Society Open Science. Authors may consider the following recommendations to improve the quality of the manuscript:

1) Introduction, Page 3, Lines 1-2: A comprehensive study demonstrating the suitability of adsorption to remove and even recover phosphorus from low concentrated secondary effluent wastewater, including a verification of the long-term reusability of the adsorbents, is recommended:

"Polishing of secondary wastewater effluents through elimination and recovery of dissolved phosphorus with reusable magnetic microsorbents". Proceedings of the Water Environment Federation (WEF): Nutrient Symposium 2017 (3), ISSN 1938-6478, 169–181, DOI: <https://doi.org/10.2175/193864717821494169>

2) Introduction, Page 3, Lines 9-10: The following very systematic screening study representing the high phosphorus adsorption and desorption efficiency of various metal (oxy)hydroxides can be added here:

"Influence of cation building blocks of metal hydroxide precipitates on their adsorption and desorption capacity for phosphate in wastewater - A screening study". Colloids and Surfaces A: Physicochem. and Eng. Aspects, 488, 145–153. DOI: <https://doi.org/10.1016/j.colsurfa.2015.10.017>

3) Page 3, Line 48: Note down also the respective concentration of the biochar in parenthesis: i.e. 1 g biochar in 500 mL KH₂PO₄ solution corresponds to 2 g/L biochar. The same applies to Page 4, Line 2: 0.08 g biochar in 40 mL KH₂PO₄ solution corresponds to 2 g/L biochar.

4) Page 4, Lines 27-28: Authors state that the fitting of the kinetic and isotherm models to the experimental data was performed with Origin 2018 using the non-linear fitting methods, as plotted in Figure 3 and Figure 5. Why didn't authors simply plot the smooth fitting curves based on the actual modeling parameters calculated from the respective model equations (Tables 3 and 4)?

5) Page 5, Lines 5-6: Authors measured that most of the metal-doped biochars had pHPZC around the neutral range, except for Fe-BC with pHPZC=2.1 and Mg-BC with pHPZC=11.3. They explained this with the high pHPZC~12 of MgO and likely due to the low pHPZC of Fe₃O₄ (magnetite). In support of this statement, I can recommend the paper by Mandel et al. where the researchers measured the pHPZC of various Mg-based oxyhydroxides which all had pHPZC~12, and they also measured the pHPZC of magnetite (Fe₃O₄) nanoparticles which showed pHPZC~2:

Mandel, K. et al. (2013). "Layered double hydroxide ion exchangers on superparamagnetic microparticles for recovery of phosphate from waste water". Journal of Materials Chemistry A 1 (5), 1840-1848, DOI: <https://doi.org/10.1039/C2TA00571A>

6) Table 1: There is a typo in the column named "Mg-C". Please, replace with "Mg-BC".

7) Figure 3: It would help the reader if authors include in the figure caption a remark, listing the reaction parameters of the kinetics experiment, i.e. initial P-concentration: 20 mg/L PO₄-P; biochar concentration 2 g/L, pH 7, 25°C, etc. The same refers to Figure 5 (isotherms) where authors can list the phosphate and biochar concentration range, contact time, pH, temperature, etc.

8) Authors performed the study only in a synthetic KH₂PO₄ solution. Nevertheless, they need to keep in mind that the biochar performance in a real wastewater or water matrix may lead to completely different results. The competing effect of other constituents (dissolved ions, organics, suspended matter, etc.) present in real water / wastewater matrices must be taken into account in future studies.

9) Have authors checked the chemical and mechanical stability of the developed metal-incorporated biochars? They need to prove that none of the metals or other precursors from the Me-BC are leached into the treated effluent, especially if they carry a phosphate ion with them, which would essentially pollute the effluent instead of purifying it.

10) How do authors imagine the practical engineering application of the metal-loaded biochars, i.e. how would the biochars be harvested / separated from the treated wastewater? The efficiency of the biochar separation will directly affect the effluent quality. If a simple sedimentation is proposed, probably there will be a need to dose flocculants to assure an efficient sedimentation process, but the flocculants may change the properties of the P-rich biochars and make them unsuitable for field applications, e.g. as soil amendment. Generally, in the future the authors need to investigate the plant availability and fertilizer value of the proposed P-loaded biochars. Often times the crystalline metal-phosphate precipitates, e.g. AlPO₄, are not suitable as fertilizers and have poor plant availability, which might defeat the original concept of this study to utilize the P-loaded biochars directly on the field!

Reviewer: 2

Comments to the Author(s)

This work actually lots of effective experimental data. Publication of these data in public open access journal media would have good contribution to the related research communities. From the latter positive point, I basically recommend publication of this manuscript in Royal Society Open Science. Please see below necessary revisions.

1) Adsorption results were well reported, but mechanisms are not fully clear. The latter matters would be revealed by appropriate spectral analyses. IR measurement of these bio-derived carbon materials (with metal oxides) with phosphates had better be measured. From the peak shifts, interaction between the materials and phosphates can be discussed.

2) References can be more generalized and updated. For example, biomass-derived carbon materials are hot subjects and related papers can be cited more (for example, see below).
<https://www.journal.csj.jp/doi/10.1246/bcsj.20200055>
<https://www.sciencedirect.com/science/article/abs/pii/S030438941931194X?via%3Dihub>

3) Please provide clear scale bars to images in Figure S1.

Author's Response to Decision Letter for (RSOS-201789.R0)

See Appendix A.

RSOS-201789.R1 (Revision)

Review form: Reviewer 1

Is the manuscript scientifically sound in its present form?

Yes

Are the interpretations and conclusions justified by the results?

Yes

Is the language acceptable?

Yes

Do you have any ethical concerns with this paper?

No

Have you any concerns about statistical analyses in this paper?

No

Recommendation?

Accept with minor revision (please list in comments)

Comments to the Author(s)

The revised manuscript RSOS-201789.R1 entitled "A comparative study on phosphate removal from water using phragmites australis biochars loaded with different metal oxides" fulfills all expected revisions supposed to be made and is now of good publishable quality. The authors have carefully addressed every single comment and remark from both reviewers. The answers are elaborate, extensive and provide enough arguments and proof to validate the authors' statements. All necessary changes have been made based on the reviewers' recommendations. In my opinion, in this revised form the manuscript should be accepted for publication in the journal Royal Society Open Science after authors take into consideration one more minor remark:

The literature reference 4 from Drenkova-Tuhtan et al. can be complemented with a recently published and more detailed full journal article on the same topic:

Drenkova-Tuhtan, A.; Sheeleigh, E.K.; Rott, E.; Meyer, C.; Sedlak, D.L. (2021). Sorption of recalcitrant phosphonates in reverse osmosis concentrates and wastewater effluents - influence of metal ions. *Water Science & Technology* (2021), 83 (4), 934-947.
<https://doi.org/10.2166/wst.2021.026>.

Review form: Reviewer 2

Is the manuscript scientifically sound in its present form?

Yes

Are the interpretations and conclusions justified by the results?

Yes

Is the language acceptable?

Yes

Do you have any ethical concerns with this paper?

No

Have you any concerns about statistical analyses in this paper?

No

Recommendation?

Accept as is

Comments to the Author(s)

Replies and revisions are fine. The revised version becomes acceptable.

Decision letter (RSOS-201789.R1)

Dear Dr Wang:

Title: A comparative study on phosphate removal from water using phragmites australis biochars loaded with different metal oxides

Manuscript ID: RSOS-201789.R1

Thank you for submitting the above manuscript to Royal Society Open Science. On behalf of the Editors and the Royal Society of Chemistry, I am pleased to inform you that your manuscript will be accepted for publication in Royal Society Open Science subject to minor revision in accordance with the referee suggestions. Please find the reviewers' comments at the end of this email.

The reviewers and handling editors have recommended publication, but also suggest some minor revisions to your manuscript. Therefore, I invite you to respond to the comments and revise your manuscript.

Because the schedule for publication is very tight, it is a condition of publication that you submit the revised version of your manuscript before 31-Mar-2021. Please note that the revision deadline will expire at 00.00am on this date. If you do not think you will be able to meet this date please let me know immediately.

When submitting your revised manuscript, you will be able to respond to the comments made by the referees and upload a file "Response to Referees" in "Section 6 - File Upload". You can use this to document any changes you make to the original manuscript. In order to expedite the

processing of the revised manuscript, please be as specific as possible in your response to the referees.

Kind regards,
Dr Laura Smith
Publishing Editor, Journals

On behalf of the Subject Editor Professor Anthony Stace and the Associate Editor Dr Nadia Martinez Villegas.

RSC Associate Editor:

Comments to the Author:

The authors have satisfactorily addressed every single comment and remark from the reviewers. Your manuscript can now be accepted with minor revisions.

RSC Associate Editor:

Comments to the Author:
(There are no comments.)

Reviewer comments to Author:
Reviewer: 2

Comments to the Author(s)
Replies and revisions are fine. The revised version becomes acceptable.

Reviewer: 1

Comments to the Author(s)
The revised manuscript RSOS-201789.R1 entitled "A comparative study on phosphate removal from water using phragmites australis biochars loaded with different metal oxides" fulfills all expected revisions supposed to be made and is now of good publishable quality. The authors have carefully addressed every single comment and remark from both reviewers. The answers are elaborate, extensive and provide enough arguments and proof to validate the authors' statements. All necessary changes have been made based on the reviewers' recommendations. In my opinion, in this revised form the manuscript should be accepted for publication in the journal Royal Society Open Science after authors take into consideration one more minor remark:

The literature reference 4 from Drenkova-Tuhtan et al. can be complemented with a recently published and more detailed full journal article on the same topic:

Drenkova-Tuhtan, A.; Sheeleigh, E.K.; Rott, E.; Meyer, C.; Sedlak, D.L. (2021). Sorption of recalcitrant phosphonates in reverse osmosis concentrates and wastewater effluents – influence of metal ions. *Water Science & Technology* (2021), 83 (4), 934-947.
<https://doi.org/10.2166/wst.2021.026>.

Author's Response to Decision Letter for (RSOS-201789.R1)

See Appendix B.

Decision letter (RSOS-201789.R2)

Dear Dr Wang:

Title: A comparative study on phosphate removal from water using phragmites australis biochars loaded with different metal oxides
Manuscript ID: RSOS-201789.R2

It is a pleasure to accept your manuscript in its current form for publication in Royal Society Open Science. The chemistry content of Royal Society Open Science is published in collaboration with the Royal Society of Chemistry.

On behalf of the Subject Editor Professor Anthony Stace and the Associate Editor Dr Nadia Martinez Villegas.

RSC Associate Editor
Comments to the Author:
The authors have addressed all necessary comments and remarks from the reviewers. The manuscript can now be accepted as is.

Reviewer(s)' Comments to Author:

Appendix A

Response to Referees

Reviewer: 1

1) Introduction, Page 3, Lines 1-2: A comprehensive study demonstrating the suitability of adsorption to remove and even recover phosphorus from low concentrated secondary effluent wastewater, including a verification of the long-term reusability of the adsorbents, is recommended:

"Polishing of secondary wastewater effluents through elimination and recovery of dissolved phosphorus with reusable magnetic microsorbents". Proceedings of the Water Environment Federation (WEF): Nutrient Symposium 2017 (3), ISSN 1938-6478, 169–181, DOI: <https://doi.org/10.2175/193864717821494169>

Response: This paper (reference 4) is cited at the end of Page 2, Lines 1-2 in the revised manuscript.

2) Introduction, Page 3, Lines 9-10: The following very systematic screening study representing the high phosphorus adsorption and desorption efficiency of various metal (oxy)hydroxides can be added here:

"Influence of cation building blocks of metal hydroxide precipitates on their adsorption and desorption capacity for phosphate in wastewater - A screening study". Colloids and Surfaces A: Physicochem. and Eng. Aspects, 488, 145–153. DOI: <https://doi.org/10.1016/j.colsurfa.2015.10.017>

Response: This paper (reference 23) is cited at the end of Page 2, Line 10 in the revised manuscript.

3) Page 3, Line 48: Note down also the respective concentration of the biochar in parenthesis: i.e. 1 g biochar in 500 mL KH₂PO₄ solution corresponds to 2 g/L biochar. The same applies to Page 4, Line 2: 0.08 g biochar in 40 mL KH₂PO₄ solution corresponds to 2 g/L biochar.

Response: The phrase "(biochar concentration 2 g/L)" is added after "1 g biochar in 500 mL KH₂PO₄ solution" and "0.08 g biochar and 40 mL KH₂PO₄ solution" to specify the concentration of the biochar used in the experiments.

4) Page 4, Lines 27-28: Authors state that the fitting of the kinetic and isotherm models to the experimental data was performed with Origin 2018 using the non-linear fitting methods, as plotted in Figure 3 and Figure 5. Why didn't authors simply plot the smooth fitting curves based on the actual modeling parameters calculated from the respective model equations (Tables 3 and 4)?

Response: In order to visually show the differences in the phosphate adsorption kinetics and isotherms of the different materials, the experimental data of the 6 biochars were plotted in one figure, i.e., Figure 3 and Figure 5 respectively. For each material, 4 kinds of kinetic models were fitted to the kinetic data, and 3 kinds of isotherm models were fitted to the isotherm data. If the fitting curves of the kinetic and isotherm models were plotted in the figure, there would be 24 and 18 fitting curves in Figure 3 and Figure 5 respectively. That would be very difficult to show all the information clearly, especially for the biochars with some of the experimental data being close or overlapping, e.g., phosphate adsorption isotherms of Fe-BC and La-BC. From these

considerations, the model fitting parameters were given separately in Table 3 and Table 4 to clearly show the model fitting results.

5) Page 5, Lines 5-6: Authors measured that most of the metal-doped biochars had pHPZC around the neutral range, except for Fe-BC with pHPZC=2.1 and Mg-BC with pHPZC=11.3. They explained this with the high pHPZC~12 of MgO and likely due to the low pHPZC of Fe₃O₄ (magnetite). In support of this statement, I can recommend the paper by Mandel et al. where the researchers measured the pHPZC of various Mg-based oxyhydroxides which all had pHPZC~12, and they also measured the pHPZC of magnetite (Fe₃O₄) nanoparticles which showed pHPZC~2:

Mandel, K. et al. (2013). "Layered double hydroxide ion exchangers on superparamagnetic microparticles for recovery of phosphate from waste water". Journal of Materials Chemistry A 1 (5), 1840-1848, DOI: <https://doi.org/10.1039/C2TA00571A>

Response: We appreciate for this suggestion. In the manuscript, the sentence "Fe-BC had a low pHPZC (2.1), likely due to the low pHPZC of Fe₃O₄." is revised to "Fe-BC had a low pHPZC (2.1) due to the low pHPZC (2.0) of Fe₃O₄." with this recommended paper (reference 34) cited at the end of this sentence.

6) Table 1: There is a typo in the column named "Mg-C". Please, replace with "Mg-BC".

Response: "Mg-C" in the first row of Table 1 is revised to "Mg-BC".

7) Figure 3: It would help the reader if authors include in the figure caption a remark, listing the reaction parameters of the kinetics experiment, i.e. initial P-concentration: 20 mg/L PO₄-P; biochar concentration 2 g/L, pH 7, 25°C, etc. The same refers to Figure 5 (isotherms) where authors can list the phosphate and biochar concentration range, contact time, pH, temperature, etc.

Response: We accept this suggestion. The experimental parameters used in the phosphate adsorption kinetic and isotherm experiments are included in the figure captions of Figure 3 and Figure 5 in parenthesis. Specifically, the figure caption of Figure 3 is revised to "Phosphate adsorption kinetics of the biochars (initial phosphate concentration 20 mg/L PO₄³⁻-P, biochar concentration 2 g/L, initial solution pH 7.0, temperature 25°C, shaking speed 150 r/min, reaction time 72 h)". The figure caption of Figure 5 is revised to "Phosphate adsorption isotherm of the biochars (initial phosphate concentration 100-3000 mg/L PO₄³⁻-P for Al-BC and 100-1500 mg/L for the other five biochars, biochar concentration 2 g/L, initial solution pH 7.0, temperature 25°C, shaking speed 150 r/min, reaction time 168 h for Al-BC and 24 h for the other biochars)".

8) Authors performed the study only in a synthetic KH₂PO₄ solution. Nevertheless, they need to keep in mind that the biochar performance in a real wastewater or water matrix may lead to completely different results. The competing effect of other constituents (dissolved ions, organics, suspended matter, etc.) present in real water / wastewater matrices must be taken into account in future studies.

Response: We totally agree with the referee that the phosphate adsorption kinetics and capacities of the biochars may change in real water/wastewater due to the competing effects of other constituents. This aspect was also mentioned in our manuscript in Page 5, Line 55-57: "The higher phosphate removal efficiency in the present study could be partly due to the absence of

interfering substances in the synthetic solution, while the diluted urine contained many other organic and inorganic substances besides phosphate which could compete for or block the adsorption sites.”. A sentence “In further studies, effects of various competing constituents in the real polluted water on the phosphate adsorption characteristics of the metal oxide loaded biochars and stability of them in the real polluted water need to be clarified before applying them in real practice.” is added at the end of the conclusion in the revised manuscript.

9) Have authors checked the chemical and mechanical stability of the developed metal-incorporated biochars? They need to prove that none of the metals or other precursors from the Me-BC are leached into the treated effluent, especially if they carry a phosphate ion with them, which would essentially pollute the effluent instead of purifying it.

Response: In preliminary experiments we measured the phosphate concentration released from the pristine biochar and metal oxide loaded biochars by placing 0.2 g biochar in 80 mL deionized water and shaking the mixture at 25°C and 200 r/min for 24 h. Negligible phosphate ions (PO_4^{3-} -P concentration smaller than the detection limit of 0.01 mg/L) existed in the solution after shaking of the mixture with the metal oxide loaded biochars, while 0.49 mg/L PO_4^{3-} -P (corresponding to a PO_4^{3-} -P release rate of 0.20 mg/g biochar) existed in the solution after shaking of the mixture with the pristine biochar. The results proved that barely any phosphate ions were released from the biochars except the pristine biochar. This can be explained by the fact that P is an essential plant growth element and contained in the biomass feedstock (reed). After pyrolysis, P in the biomass is retained in the biochar. When the pristine biochar is placed in the aqueous solution, a part of P is dissolved and released in the solution. When preparing the metal oxide loaded biochars, the biomass is impregnated with metal chlorides. During pyrolysis, P in the biomass reacts with the metal ions and transforms into phosphate compounds with a low solubility. Through such mechanisms, P in the biomass is fixed in the metal oxide loaded biochars and negligible phosphate ions can be released in the solution. This explanation is also supported by the literature. For example, Jiang et al. [40] found that phosphates in the poultry litter transformed into less soluble forms like farringtonite ($\text{Mg}_3(\text{PO}_4)_2$) when converting the poultry litter to biochar at pyrolysis temperature above 500°C and the water extractable P decreased significantly.

To make this point clear in the manuscript, several sentences are added in Page 4, Line 49-54: “In the preliminary experiments where 0.2 g of the six biochars were placed in 80 mL deionized water respectively and shaken at 25°C and 200 r/min for 24 h, release of phosphate from the BC to the deionized water was also observed since P in the reed was retained in the BC. However, negligible phosphates were released from the five modified biochars to the deionized water. Labile phosphates in the reed transformed to less soluble forms during the pyrolysis of the metal chlorides impregnated reed by binding with the metal ions [40].”

The release of other substances, e.g., dissolved organic matter and metal ions, from the metal oxide loaded biochars to the solution was not examined, because the focus of the present study is phosphate. However, we agree with the referee that the stability of the material is important and needs further research. This point is added in the last sentence of the conclusion part of the revised manuscript: “In further studies, effects of various competing constituents in the real

polluted water on the phosphate adsorption characteristics of the metal oxide loaded biochars and stability of the biochars in the real polluted water need to be clarified before applying them in real practice.”.

10) How do authors imagine the practical engineering application of the metal-loaded biochars, i.e. how would the biochars be harvested / separated from the treated wastewater? The efficiency of the biochar separation will directly affect the effluent quality. If a simple sedimentation is proposed, probably there will be a need to dose flocculants to assure an efficient sedimentation process, but the flocculants may change the properties of the P-rich biochars and make them unsuitable for field applications, e.g. as soil amendment. Generally, in the future the authors need to investigate the plant availability and fertilizer value of the proposed P-loaded biochars. Often times the crystalline metal-phosphate precipitates, e.g. $AlPO_4$, are not suitable as fertilizers and have poor plant availability, which might defeat the original concept of this study to utilize the P-loaded biochars directly on the field!

Response: Biochar particles in the size range of 0.25-0.50 mm were used in the present study to adsorb phosphate from the solution. These particles could quickly settle after shaking, i.e., simple sedimentation was able to separate the biochar particles from the solution. Considering that a certain kind of more efficient mixing, e.g., with a mechanical mixer or effluent circulation, of the biochar particles and the to-be treated water/wastewater is to be used in practical applications, some of the biochar particles might be fragmented and the sedimentation efficiency might deteriorate. Additional operations such as letting the treated effluent pass through sieves with small mesh sizes can be applied. With such methods, the dose of flocculants can be avoided.

Previous research [7-9] has proved that the P-adsorbed MgO loaded biochar promotes the seed germination and growth of plants, confirming that the P-adsorbed MgO loaded biochar can be used as a slow-release fertiliser to improve soil quality and productivity. One conclusion of the present study is that Mg-BC is suggested for phosphate removal from water considering both phosphate adsorption rate and capacity. Therefore, the original concept of this study to utilize the P-loaded biochars on the field as a slow-release fertiliser is realistic. Knowledge of the applicability of P-adsorbed Al-BC, Ca-BC, Fe-BC and La-BC as slow-release fertilisers is however lacking, which will be investigated in our future research. In the manuscript, the sentence “Moreover, the P-adsorbed biochar can be reused as a soil amendment to improve soil quality, enabling simultaneous P recovery and waste biomass reuse.” (Page 2, Line 4-6) is changed to “Moreover, the P-adsorbed biochar can be reused as soil amendment and slow-release fertiliser to improve soil quality and productivity, enabling simultaneous P recovery and waste biomass reuse.”, and the reference at the end of this sentence is changed to [7-9] to be more specific.

Reviewer: 2

1) Adsorption results were well reported, but mechanisms are not fully clear. The latter matters would be revealed by appropriate spectral analyses. IR measurement of these bio-derived carbon materials (with metal oxides) with phosphates had better be measured. From the peak shifts, interaction between the materials and phosphates can be discussed.

Response: FTIR analysis of the six biochars after phosphate adsorption has been additional conducted. The FTIR spectra of each biochar before and after phosphate adsorption are shown in Figure 1 of the revised manuscript. Comparison of the peak intensities before and after adsorption reveals that new peaks in the ranges of 1157-1029/cm and 616-563/cm, corresponding to the P-O stretching and O-P-O bending vibrations respectively, appeared on the six biochars after adsorption. The Al-O, La-O and Mg-O peaks of the Al-BC, La-BC and Mg-BC respectively weakened after adsorption while the -OH peaks of them strengthened, which could be explained by the transformation of some Al, La and Mg oxides into hydroxyl compounds in the solution through hydrolysis reactions. For the Ca-BC and Fe-BC, the -OH peaks of the two biochars and the Fe-O peak of Fe-BC weakened, indicating the involvement of the metal-O and -OH groups in the adsorption process. These results indicate that when interacting with the phosphate solution, the metal-O and -OH groups of the biochars bound or exchanged with phosphate, forming mono-, bi- and tri-nuclear inner-sphere surface complexes, thus phosphate ions were adsorbed. The text in the manuscript (Page 6, Line 54-Page 7, Line 4) has been revised accordingly.

2) References can be more generalized and updated. For example, biomass-derived carbon materials are hot subjects and related papers can be cited more (for example, see below).

<https://www.journal.csj.jp/doi/10.1246/bcsj.20200055>

<https://www.sciencedirect.com/science/article/abs/pii/S030438941931194X?via%3Dihub>

Response: References [5] and [6] of the manuscript are changed to the above two papers suggested by the referee.

3) Please provide clear scale bars to images in Figure S1.

Response: Figure S1 is revised by adding clear scale bars to the images.

Appendix B

Response to Referees

Reviewer: 1

The literature reference 4 from Drenkova-Tuhtan et al. can be complemented with a recently published and more detailed full journal article on the same topic:

Drenkova-Tuhtan, A.; Sheeleigh, E.K.; Rott, E.; Meyer, C.; Sedlak, D.L. (2021). Sorption of recalcitrant phosphonates in reverse osmosis concentrates and wastewater effluents – influence of metal ions. *Water Science & Technology* (2021), 83 (4), 934-947. <https://doi.org/10.2166/wst.2021.026>.

Response: Reference 4 of the manuscript has been substituted with the recommended paper.